# A developmental analysis of dimensions of empathy during early adolescence: Behavioral empathy but not cognitive empathy is associated with lower psychopathology

**Megan Cherewick**[1]*, **Sarah Schmiege**[2], **Emily Hipp**[2], **Jenn Leiferman**[1], **Prosper Njau**[3], **Ronald E. Dahl**[4]

**1** Department of Community & Behavioral Health, Colorado School of Public Health, Aurora, CO, United States of America, **2** Department of Biostatistics & Informatics, Colorado School of Public Health, Aurora, CO, United States of America, **3** Health for a Prosperous Nation, Dar es Salaam, Tanzania, **4** Institute of Human Development, University of California Berkeley, Berkeley, CA, United States of America

* megan.cherewick@cuanschutz.edu

**Data Availability Statement:** The raw data supporting the conclusions of this article are made

## Abstract

Construct definitions of empathy have sought to distinguish between different dimensions of empathetic capacity that are significantly associated with psychological distress or wellbeing. Research has provided substantial evidence differentiating affective and cognitive empathy; however, more recent research has cited the importance of a third domain represented by empathetic behaviors and compassionate intent to comfort others. Examining developmental and maturational stage during the rapid transitional period of early adolescence is needed to model developmental trajectories, mechanisms of change and mental health outcomes. This study aims to assess relationships between pubertal developmental stage, dimensions of empathy, and depression, anxiety and externalizing behaviors among early adolescents. A cross-sectional survey among young adolescents ages 9–12 years was conducted in Dar es Salaam, Tanzania. The relationships between pubertal developmental stage, affective, cognitive, and behavioral empathy scores and internalizing and externalizing symptoms were evaluated using hierarchical regression modeling. Structural equation modeling was used to test a theoretical model of structural paths between these variables. A sample of 579 very young adolescents (270 boys and 309 girls) aged 9–12 years participated in the study. Pubertal development scale scores were associated with affective, cognitive, and behavioral empathy. Adolescents who had greater behavioral empathy scores reported lower internalizing and externalizing symptoms. Adolescents who had transitioned further through puberty and had higher cognitive empathy scores and reported higher internalizing symptoms. These findings support the importance of pubertal developmental stage in assessing risk and protective paths to mental health during adolescence. While empathetic capacity is widely perceived to be a positive trait, dimensional analysis of empathy among early adolescents indicates that behavioral skills and compassionate acts may be particularly protective and promote positive mental health outcomes.

available by the authors without undue reservation as Supporting Information (S1 Dataset).

**Funding:** This publication is based on research funded by the Bill and Melinda Gates Foundation (OPP1158584 to RD). The funders had no role in study design, data collection and analysis, decision to publish, or preparation of the manuscript. Findings and conclusions contained within are those of the authors and do not reflect positions or policies of the Bill and Melinda Gates Foundation.

**Competing interests:** The authors have declared that no competing interests exist.

## Introduction

Empathy has been defined as a character trait representing the affective and moral capacity to understand emotional states of others [1]. Researchers have also sought to distinguish between sympathy, empathy, and compassion and more recent research has sought to differentiate dimensions of empathy. A review of empathy measures by Sesso et al., 2021, found that most empathy measures have been shown to be multi-dimensional with affective and cognitive sub-dimensions most often cited in validation studies [2]. Sympathy has been equated with affective empathy, an emotionally reactive response from the individual observing distress, while cognitive empathy is the capacity to intellectually understand another person's experience, appraise and identify with emotional states in ways that goes beyond the threshold of affective empathy or sympathy [3–6].

As a latent construct, empathy has been shown to be significantly associated with prosocial behavior [7] and externalizing problems, such as aggression and substance misuse [8, 9]. Studies exploring dimensions of empathy and relationships with psychopathology have found that cognitive empathy predicts higher friendship quality [10–12]. More recently, studies of empathy have considered a third dimension of empathy in addition to affective and cognitive empathy. This dimension, 'intention to comfort' represents compassionate actions and differentiates from both affective and cognitive empathy to include behavioral empathy, the conscious decision making to motivate practice of a prosocial action [13]. This third dimension of behavioral empathy is important because cognitive empathy alone, without the skills to alleviate distress may correlate with co-rumination, the tendency to frequently discuss problems with peers that has been identified as a risk factor for depression during adolescence [14]. Compassionate behavior can be practiced externally in response to another person's distress or internally to alleviate one's own distress. Self-compassion has gained increasing attention as a protective asset that can function at the intrapersonal level [15]. In other studies, empathy explained sample variance in mental health outcomes after adjustment for coping struggles and social support [16]. Similarly, research has also recognized that individuals vary in their capacity for *positive empathy*, the capacity to share joy and happiness experienced by others [17].

Empathy research is motivated by growing evidence that empathy is associated with both mental health and wellbeing outcomes [7, 8, 18]. In adolescent research, capacity for empathy has been considered integral to social and moral development [2]. Higher levels of empathy have been associated with prosocial behaviors, higher levels of emotional regulation, and lower aggressive behaviors [19–21]. Evidence suggests the important role empathy plays in other kinds of socially undesirable behaviors such as violence, bullying, and sexual assault perpetration [22, 23]. Research has found that lower empathy is associated with attention deficit and hyperactivity disorder (ADHD) and eating disorders [24]. As a protective asset, affective empathy is associated with emotional regulation in social situations and cooperation towards collective goals [25].

The developmental science of adolescence seeks to enhance precision with which we understand the dynamic maturational period from the onset of puberty to adulthood. Much research to date has focused on differentiating how dimensional factors that comprise empathy explain risk for psychiatric disorders including autism spectrum disorder, ADHD, callous unemotional traits, and aggressive behaviors [26–30]. Developmental neuroscience suggests that dimensions of empathy are explained by different brain systems [31]. For example, development of prefrontal neural circuitry may facilitate empathic responses through higher working memory required to appraise others affective states [32]. Appraisal of other's distress is tied to several other parts of the brain that are activated during appraisal [31].

Differential associations by gender have important implications for understanding the dimensional structure of empathy. Research indicates that girls have shown higher overall empathetic capacity than boys [33, 34]. In one study, as girls increased in age affective and cognitive empathy increased, whereas in boys, affective, cognitive, and behavioral empathy were negatively associated with age [13]. Interaction effects between developmental age and gender on empathy are plausible and important because girls advance through the pubertal transition at an earlier age than boys on average [35].

Understanding the complex, dynamic, developmental trajectories of the adolescent brain allows a more complete understanding of how to match motivational proclivities and sensitive periods for learning that can shape health trajectories in the near- and long-term. Currently, little is known about high impact developmental periods to promote empathetic capacity. Understanding the significance of empathy during different maturational periods of mental health can inform practice and policy with greater precision and effect.

## Study objectives

The primary objective of this study was to evaluate theoretical relationships between pubertal development stage, mediated by dimensions of empathy (affective, cognitive, and behavioral) on internalizing symptoms (e.g., depression and anxiety) and externalizing symptoms (e.g., aggression and substance use) during early adolescence.

## Materials and methods

### Ethics statement

This study was approved by the University of California Berkeley Committee for Protection of Human Subjects Institutional Review Board (IRB)—(CPHS Protocol Number: 2018-01-10628); in June 2018. The primary local partner in Tanzania, Health for a Prosperous Nation, obtained ethical clearance for these research activities from the National Institute of Medical Research–the local IRB in Tanzania (Ref. NIMR/HQ/R.8a/Vol. IX/2851) in August 2018. All parents/caregivers provided written consent and all adolescents provided verbal assent prior to administration of the survey questionnaire. Due to limited literacy of adolescents, verbal assent was provided and is typical procedural practice in research studies including children in this context.

### Study procedures and sample selection

Participants in this study were recruited from the peri-urban Temeke Municipality in Dar es Salaam, Tanzania to participate in a three-arm comparative effectiveness trial Discover, an intervention to support social emotional mindsets and skills among very young adolescents. A detailed study protocol of the parent study, Discover Learning, is provided elsewhere, including additional details on the subsequent three-arm comparative effectiveness trial and participant recruitment and eligibility criteria [36]. This trial explored whether experiential, social emotional learning for early adolescents ages 10–12, affected mental health and wellbeing outcomes. Data were collected at baseline prior to the start of the intervention in June-July 2019, and presents a cross-sectional view of associations between empathy, pubertal development stage, and internalizing/externalizing symptoms prior to randomization. The analytic sample for this study was comprised of 579 adolescents (270 boys and 309 girls) ages 9–12 (mean age = 10.48; $SD$ = 0.55).

## Survey measures

All participants completed a survey at baseline including measures of demographic characteristics (e.g., pubertal development, gender, age), social emotional skills and mindsets (i.e., affective, cognitive and behavioral empathy) and psychosocial assessment (i.e., depression, anxiety and externalizing behaviors). Each measurement scale was selected based on previous use in low- and middle-income countries with adolescent populations. Age and sex were included as covariates in this study.

## Pubertal Development Scale (PDS)

Self-reported pubertal development was measured using the Pubertal Development Scale (PDS) [37], a scale that has been validated and correlated with Tanner staging methods [38]. The PDS has a reported alpha of 0.67–0.70 and includes different questions for boys and girls such as, "Have you noticed a deepening of your voice?" and "Have you begun to grow hair on your face?" for boys; "Have you noticed that your breasts have begun to grow?" and "Have you begun to menstruate?" for girls [37]. Participants responded to three items about perceived changes in maturation on a dichotomous scale. Average ratings were computed so that higher scores indicated higher pubertal development.

## Empathy

Empathy was measured with the Empathy Questionnaire for Children and Adolescents (EmQue-CA) scale [13]. This scale has been validated with Tanzanian adolescents to assess affective empathy, cognitive empathy, and intention to comfort [39]. Affective empathy measures the extent to which adolescents feel the emotional state of the suffering person and includes five items. Cognitive empathy measures the extent to which the adolescent understands why another person is in distress and includes three items. Intention to comfort includes five items that measure the extent to which the adolescent is inclined to actively help or support the suffering person [13]. We refer to the intention to comfort factor as *behavioral* empathy in this study, as it captures a unique third dimension of empathy that highlights not just an understanding of another person's distress, but how a person intends to behave and react based on that understanding. Response categories asked participants to rate on a 3-point scale whether the statement was 1 = "Not true", 2 = "Somewhat true" or 3 = "True". Mean scores were calculated for each subscale such that higher scores reflect higher empathy in that domain. Previous validation of this scale demonstrated adequate internal consistency for cognitive empathy $(\alpha = 0.89)$, affective empathy $(\alpha = 0.65)$, and behavioral empathy $(\alpha = 0.75)$ [39].

## African Youth Psychological Assessment (AYPA)

The African Youth Psychological Assessment (AYPA), measures internalizing and externalizing symptoms in adolescents and has been used previously in Tanzania with minor adaptations [40, 41]. A study to validate the AYPA using a comparison of parent and self-reported ratings on presence of local syndrome terms have been published previously [40]. Among the 166 youth in the validity study, the AYPA demonstrated satisfactory internal reliability, ranging from $\alpha = 0.70$–0.87 for each subscale and further validated subdimensions of depression and anxiety within the internalizing symptom item measures [42]. Table 1 summarizes the items and subscales in this study's adaptation of the AYPA. Participants were asked each item of the AYPA on a 4-point scale of 1 = "Never", 2 = "Somewhat", 3 = "Often", 4 = "All the time". The Cronbach's $\alpha$ for this scale lies between 0.72 and 0.88 [40]. All sub-scales had

**Table 1. Sample characteristics (*N* = 579).**

| | *N* | % |
|---|---|---|
| Gender | | |
| Boys | 270 | 46.6 |
| Girls | 309 | 53.4 |
| Age | | |
| 9 | 13 | 2.3 |
| 10 | 276 | 47.7 |
| 11 | 288 | 49.7 |
| 12 | 2 | 0.4 |
| Mean Age (*SD*) | 10.48 (0.55) | |
| Grade | | |
| 3 | 172 | 29.7 |
| 4 | 261 | 45.1 |
| 5 | 146 | 25.2 |
| Live with both parents | | |
| No | 173 | 33.4 |
| Yes | 345 | 66.6 |
| Household profile | Mean | *SD* |
| Household Size | 5.7 | 2.6 |
| Tanzanian Poverty Score (0–71)[1] | 61.4 | 11.6 |

Note.

[1]Summative score of items 2–10 of the Tanzanian Poverty Scorecard; Higher score = higher wealth

satisfactory to excellent alpha values in this sample, listed as follows: prosocial/adaptive ($\alpha$ = 0.72), somatic complaints without medical cause ($\alpha$ = 0.74), externalizing problems ($\alpha$ = 0.83), and internalizing problems ($\alpha$ = 0.88) [40].

## Data analysis

Data was analyzed using Stata Statistical Software [43]. Sample characteristics are presented using frequencies and means by gender. All key study variables were assessed for significant correlational relationship with variables using Pearson's correlations. Hierarchical robust regression models were fitted to examine the association of affective, cognitive, and behavioral empathy on psychological symptom measures (depression, anxiety, and externalizing behaviors). Pubertal development status and sex were included as covariates in Model 1. Model 2 added dimensions of empathy (affective, cognitive, and behavioral). Model 3 added the significant interaction term of pubertal development status and cognitive empathy.

Structural equation modeling (SEM) was completed to estimate paths from pubertal development status to affective, cognitive, and behavioral empathy; and from each empathy subdimension to psychological symptoms. The structural path model is consistent with previously developed conceptual models and theory identifying paths from pubertal development to empathetic capacity and associated relationships with mental health outcomes. The maximum likelihood method of estimation was used to estimate the SEM and was assessed for goodness of fit. The following statistical criteria was used to evaluate model fit: $RMSEA < 0.06$; $CFI > 0.90$, $TLI > 0.90$, and $SRMR < 0.08$ [44].

## Results

### Sample characteristics

Five hundred and seventy-nine (579) adolescents ages 9–12 were included in the analytic sample of this study (Table 1). The sample included 270 boys (46.6%) and 309 girls (53.4%). The mean age of study participants was 10.48 ($SD$ = 0.55). 172 (29.7%) of participants were in 3rd grade, 261 (45.1%) in 4th grade and 146 (25.2%) in 5th grade. 345 (66.6%) of participants lived with both parents and 173 (33.4%) did not live with both parents. The average size of the household of study participants was 5.7 ($SD$ = 2.6). Study participants reported a mean score of 61.4 ($SD$ = 11.6) on the Tanzanian Poverty Scorecard (range 0–71). At the time of data collection, most recent World Bank estimates reported a basic needs poverty line of TZS 49,320 (approximately USD 21) per adult per month [45]. In 2018, the poverty rate in Tanzania was about 26.4%, and about 49% of the population lived below the international poverty line of USD 1.90 per person per day [45]. Covariates that were not found to be associated with empathy dimensions or psychological outcomes were excluded from subsequent analyses.

Several variables included in this study were significantly correlated although the magnitude of correlation was relatively small (Table 2). Affective empathy was positively associated with female gender ($r$ = 0.11; $p < 0.05$) and pubertal development ($r$ = 0.10; $p < 0.05$). Behavioral empathy was positively correlated with pubertal development ($r$ = 0.09; $p < 0.05$); affective empathy ($r$ = 0.31; $p < 0.001$) and cognitive empathy ($r$ = 0.25; $p < 0.001$). Depression ($r$ = 0.09; $p < 0.05$); and anxiety ($r$ = 0.17; $p < 0.001$) were positively correlated with cognitive empathy. Externalizing behaviors were negatively associated with behavioral empathy ($r$ = -0.18; $p < 0.001$) and positively associated with depression ($r$ = 0.51; $p < 0.001$) and anxiety ($r$ = 0.43; $p < 0.001$).

Simple linear regressions on empathy dimension were completed by gender, age, and pubertal development stage (1–4) (Table 3). Results indicate that total empathy was higher in PDS stage 3 compared to the reference category of PDS stage 1 ($\beta$ = 0.24; $p$ = 0.024). Female gender was associated with higher affective empathy ($\beta$ = 0.21; $p$ = 0.011). PDS stage 3 ($\beta$ = 0.24; $p$ = 0.024) and PDS stage 4 ($\beta$ = 0.83; $p$ = 0.030) were significantly associated with higher affective empathy. PDS stage 4 was associated with higher scores for cognitive empathy ($\beta$ = 0.09; $p$ = 0.021).

**Table 2. Pearson correlation coefficients for key analytic variables.**

| Key Variable | 1 | 2 | 3 | 4 | 5 | 6 | 7 | 8 | 9 |
|---|---|---|---|---|---|---|---|---|---|
| 1. Sex | - | | | | | | | | |
| 2. Age | 0.06 | - | | | | | | | |
| 3. PDS | 0.03 | 0.01 | - | | | | | | |
| 4. Affective | 0.11* | -0.06 | 0.10* | - | | | | | |
| 5. Cognitive | -0.06 | 0.08 | 0.01 | 0.25*** | - | | | | |
| 6. Behavioral | 0.02 | 0.04 | 0.09* | 0.31*** | 0.08* | - | | | |
| 7. Depression | 0.01 | 0.01 | 0.07 | -0.03 | 0.09* | -0.15 | - | | |
| 8. Anxiety | 0.01 | 0.04 | 0.07 | 0.05 | 0.17*** | -0.14 | 0.75 | - | |
| 9.Externalizing Behaviors | -0.03 | 0.03 | -0.02 | -0.04 | -0.03 | -0.18*** | 0.51*** | 0.43*** | - |

Note.

*$p < 0.05$

**$p < 0.01$

***$p < 0.001$

**Table 3. Simple linear regressions by sex, age and PDS stage on empathy.**

|  | Affective | | | Cognitive | | | Behavioral | | | Total | | |
|---|---|---|---|---|---|---|---|---|---|---|---|---|
|  | *β* | SE | *p* | *β* | SE | *p* | *β* | SE | *p* | *β* | SE | *p* |
| Sex | 0.21 | 0.08 | 0.011* | -0.12 | 0.08 | 0.151 | 0.04 | 0.08 | 0.609 | 0.08 | 0.08 | 0.343 |
| Age | -0.11 | 0.08 | 0.167 | 0.14 | 0.08 | 0.067 | 0.07 | 0.08 | 0.338 | 0.01 | 0.08 | 0.864 |
| PDS 2 | -0.03 | 0.10 | 0.779 | -0.16 | 0.10 | 0.111 | 0.18 | 0.10 | 0.067 | -0.02 | 0.10 | 0.863 |
| PDS 3 | 0.24 | 0.11 | 0.024* | 0.05 | 0.11 | 0.645 | 0.21 | 0.11 | 0.052 | 0.24 | 0.11 | 0.024* |
| PDS 4 | 0.83 | 0.38 | 0.030* | 0.09 | 0.38 | 0.021* | 0.29 | 0.38 | 0.446 | 0.68 | 0.38 | 0.074 |

Note. PDS stage 1 is the reference category

*$p < 0.05$

**$p < 0.01$

***$p < 0.001$

The results of hierarchical regression analyses on depression, anxiety and externalizing behaviors are presented in Table 4. After models were fitted, we examined the variance inflation factor for all regression results to check for multicollinearity. Variance inflation factors (VIF) ranged from 1.02 to 2.79, well below the conventional cut off indicating regression results are robust to multicollinearity of VIF less than or equal to 10. Model 1 included sex, age and PDS; these variables were not significantly associated with depression, anxiety, or externalizing behaviors. Model 2 included sex, age, PDS and affective, cognitive, and behavioral

**Table 4. Hierarchical multivariable regressions on psychological symptoms.**

|  |  | Depression [t] | | | Anxiety [ç] | | | Externalizing Behaviors[Ø] | | |
|---|---|---|---|---|---|---|---|---|---|---|
| Model |  | *β* | SE | *p* | *β* | SE | *p* | *β* | SE | *p* |
| M1 | Sex | 0.06 | 0.43 | 0.884 | 0.01 | 0.16 | 0.962 | -0.13 | 0.18 | 0.474 |
|  | Age | 0.09 | 0.39 | 0.819 | 0.13 | 0.15 | 0.373 | 0.12 | 0.16 | 0.453 |
|  | PDS | 1.28 | 0.78 | 0.102 | 0.52 | 0.29 | 0.078 | -0.13 | 0.32 | 0.682 |
| M2 | Sex | 0.17 | 0.43 | 0.688 | 0.04 | 0.16 | 0.795 | -0.14 | 0.18 | 0.436 |
|  | Age | 0.06 | 0.39 | 0.872 | 0.12 | 0.14 | 0.417 | 0.17 | 0.16 | 0.301 |
|  | PDS | 1.56 | 0.77 | 0.044* | 0.58 | 0.29 | 0.042* | -0.03 | 0.32 | 0.932 |
|  | Affective | -0.03 | 0.10 | 0.780 | 0.04 | 0.04 | 0.262 | 0.03 | 0.04 | 0.430 |
|  | Cognitive | 0.25 | 0.10 | 0.016* | 0.16 | 0.04 | 0.000*** | -0.03 | 0.04 | 0.452 |
|  | Behavioral | -0.47 | 0.12 | <0.001*** | -0.19 | 0.05 | <0.001*** | -0.22 | 0.05 | <0.001*** |
| M3 | Sex | 0.10 | 0.43 | 0.820 | 0.02 | 0.16 | 0.877 | -0.16 | 0.18 | 0.381 |
|  | Age | 0.09 | 0.39 | 0.827 | 0.12 | 0.14 | 0.397 | 0.17 | 0.16 | 0.286 |
|  | PDS | -0.13 | 1.06 | 0.901 | 0.21 | 0.39 | 0.598 | -0.43 | 0.44 | 0.335 |
|  | Affective | -0.03 | 0.09 | 0.743 | 0.04 | 0.04 | 0.274 | 0.03 | 0.04 | 0.446 |
|  | Cognitive | 0.03 | 0.14 | 0.808 | 0.11 | 0.05 | 0.031* | -0.08 | 0.06 | 0.151 |
|  | Behavioral | -0.46 | 0.12 | <0.001*** | -0.19 | 0.05 | <0.001*** | -0.22 | 0.05 | <0.001*** |
|  | PDS*Cognitive | 0.28 | 0.12 | 0.020* | 0.06 | 0.04 | 0.164 | 0.07 | 0.05 | 0.189 |

Note. SE = Robust standard errors

*$p < 0.05$

**$p < 0.01$

***$p < 0.001$

t $R^2 = 0.0048$ for step 1, $p = 0.4277$; $R^2 = 0.0395$ for step 2, $p = 0.0001$; $R^2 = 0.0486$ for step 3, $p = 0.0200$

ç $R^2 = 0.0068$ for step 1, $p = 0.2674$; $R^2 = 0.0648$ for step 2, $p < 0.0001$; $R^2 = 0.0680$ for step 3, $p = 0.1635$

Ø $R^2 = 0.0021$ for step 1, $p = 0.7534$; $R^2 = 0.0348$ for step 2, $p = 0.0003$; $R^2 = 0.0377$ for step 3, $p = 0.1889$

empathy. Mean PDS was associated with higher depressive symptoms ($\beta$ = 1.56; $p$ = 0.044), higher anxiety symptoms ($\beta$ = 0.25; $p$ = 0.016), and anxiety ($\beta$ = 0.16; $p < 0.001$). In contrast, behavioral empathy was associated with lower depression symptoms ($\beta$ = -0.47; $p < 0.001$) lower anxiety symptoms ($\beta$ = -0.19; $p < 0.001$) and lower reports of externalizing behaviors ($\beta$ = -0.22; $p < 0.001$). Interaction terms between PDS and empathy dimensions and PDS and sex were tested and found to be insignificant. The interaction term between PDS and cognitive empathy was significant and is included in Model 3. Model 3 results demonstrated the interaction between PDS greater than 0 and higher cognitive empathy dimension scores were associated with higher depression symptoms ($\beta$ = 0.28; $p$ = 0.020), and PDS and cognitive empathy variables were no longer significant for depression systems. This indicated that the interaction term explained relatively more variance in the outcome after adjusting for PDS and cognitive empathy variables. In Model 3, behavioral empathy remained significant with similar effect size ($\beta$ = -0.46; $p < 0.001$) in association with lower depression, anxiety, and externalizing behaviors.

## Structural Equation Model (SEM)

Structural paths between pubertal development mean score, affective, cognitive, and behavioral empathy factor scores and depression, anxiety and externalizing symptoms were tested in a structural equation model to evaluate significance of associations by empathy dimension (Fig 1). Goodness of fit indices indicated excellent model fit ($\chi^2$, df:3 = 6.656; $p < 0.001$; $CFI$ = 0.995; $TLI$ = 0.968; $SRMR$ = 0.020; $RMSEA$ = 0.046; $AIC$ = 15404.69; $BIC$ = 15535.485). Standardized path coefficients, standard errors, p-values and 95% confidence intervals are

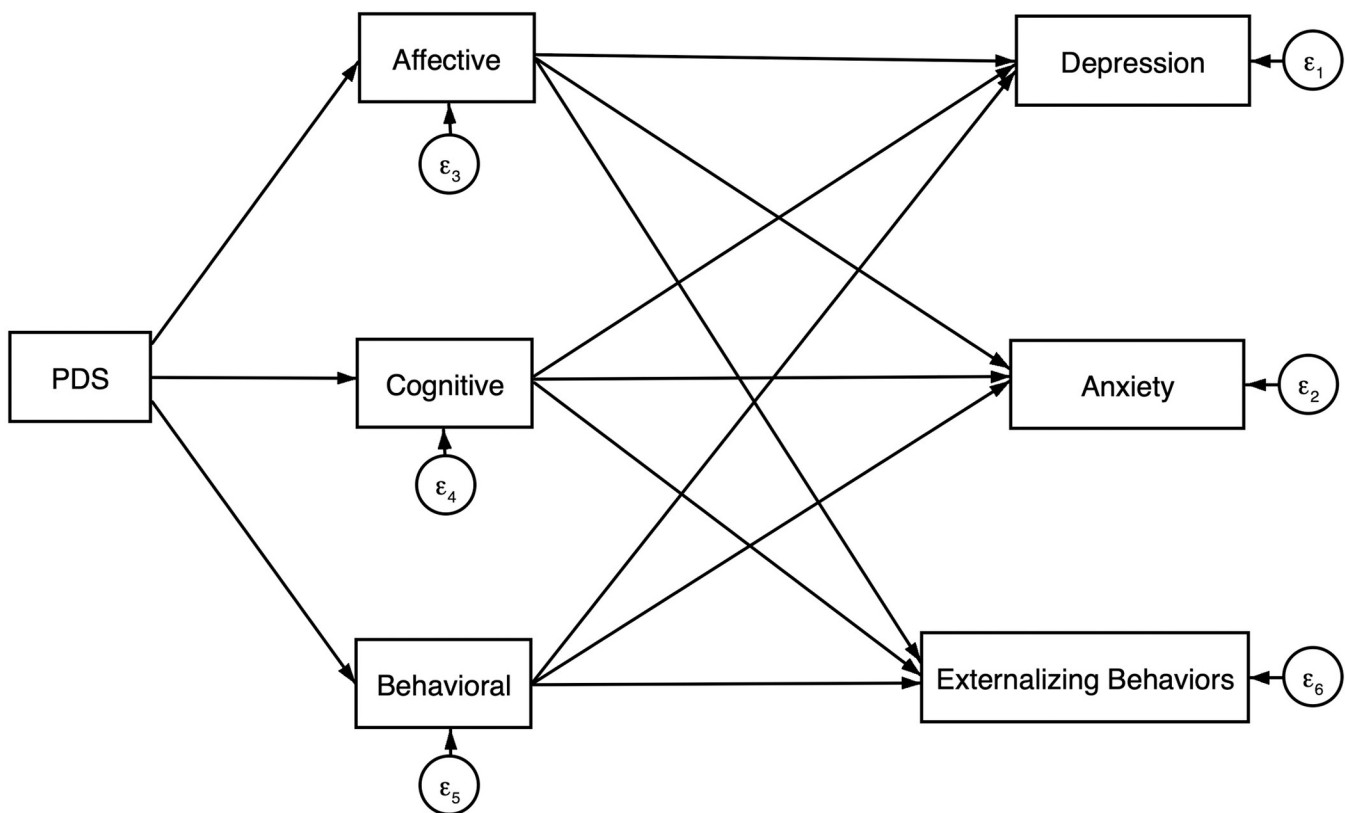

**Fig 1. Developmental structural equation model of empathy and psychological symptoms.**

**Table 5. Standardized path coefficients between risk and protective factors and psychological symptoms.**

| Structural Paths | Coefficient | SE | Z | p>|z| | 95% Confidence Interval | |
|---|---|---|---|---|---|---|
| **Affective Empathy** | | | | | | |
| PDS | 0.10 | 0.04 | 2.52 | 0.012* | 0.02 | 0.18 |
| **Cognitive Empathy** | | | | | | |
| PDS | 0.01 | 0.04 | 0.15 | 0.880 | -0.08 | 0.09 |
| **Behavioral Empathy** | | | | | | |
| PDS | 0.09 | 0.04 | 2.23 | 0.025* | 0.01 | 0.17 |
| **Depression** | | | | | | |
| Affective Empathy | 0.00 | 0.04 | -0.08 | 0.932 | -0.09 | 0.08 |
| Cognitive Empathy | 0.10 | 0.04 | 2.38 | 0.017* | 0.02 | 0.18 |
| Behavioral Empathy | -0.16 | 0.04 | -3.68 | <0.001*** | -0.24 | -0.07 |
| **Anxiety** | | | | | | |
| Affective Empathy | 0.06 | 0.04 | 1.27 | 0.205 | -0.03 | 0.14 |
| Cognitive Empathy | 0.18 | 0.04 | 4.26 | <0.001*** | 0.09 | 0.26 |
| Behavioral Empathy | -0.17 | 0.04 | -4.02 | <0.001*** | -0.25 | -0.09 |
| **Externalizing Behaviors** | | | | | | |
| Affective Empathy | 0.03 | 0.04 | 0.61 | 0.545 | -0.06 | 0.11 |
| Cognitive Empathy | -0.03 | 0.04 | -0.60 | 0.551 | -0.11 | 0.06 |
| Behavioral Empathy | -0.18 | 0.04 | -4.31 | <0.001*** | -0.27 | -0.10 |
| **Covariance** | | | | | | |
| Affective, Cognitive | 0.35 | 0.04 | 6.54 | <0.001*** | 0.18 | 0.33 |
| Affective, Behavioral | 0.30 | 0.04 | 8.05 | <0.001*** | 0.23 | 0.38 |
| Cognitive, Behavioral | 0.08 | 0.04 | 1.94 | 0.052 | 0.00 | 0.16 |
| Depression, Anxiety | 0.75 | 0.02 | 40.79 | <0.001*** | 0.71 | 0.78 |
| Depression, Externalizing | 0.50 | 0.03 | 15.90 | <0.001*** | 0.44 | 0.56 |
| Anxiety Externalizing | 0.43 | 0.03 | 12.56 | <0.001*** | 0.36 | 0.49 |

Note. $\chi^2 = 6.656$; df(3); $p < 0.001$; CFI = 0.995; TLI = 0.968; SRMR = 0.020; RMSEA = 0.046; AIC = 15404.69; BIC = 15535.485

listed in Table 5. PDS was associated with higher reported levels of affective ($\beta = 0.10$; $p = 0.012$) and behavioral ($\beta = 0.09$; $p = 0.025$) empathy scales, but not cognitive empathy ($\beta = 0.01$; $p = 0.880$). Higher reported cognitive empathy was associated with more depression ($\beta = 0.10$; $p = 0.017$) and anxiety symptoms ($\beta = 0.18$; $p < 0.001$). Behavioral empathy was associated with lower depression ($\beta = -0.16$; $p < 0.001$), anxiety ($\beta = -0.17$; $p < 0.001$) and externalizing behaviors ($\beta = -0.18$; $p < 0.001$). Affective empathy was not associated with any modeled outcomes; however, covariance paths were significant between affective and cognitive ($s = 0.35$; $p < 0.001$), and affective and behavioral ($s = 0.30$; $p < 0.001$) dimensions. The structural equation model is presented in Fig 2 with significant paths only (Fig 2).

## Discussion

The primary aim of this study was to evaluate pubertal development status, dimensional constructs of empathy and associations with psychological symptoms in early adolescents. Cognitive empathy, or the ability to intellectually feel the distress of another person was associated with higher depressive and anxiety symptoms and may be related to co-rumination, a risk factor for internalizing and externalizing problems [46–48]. While co-rumination was not measured in this study, cognitive empathy is plausibly related to co-rumination, the tendency to frequently discuss and rehash problems with peers, because of the capacity to intellectually feel

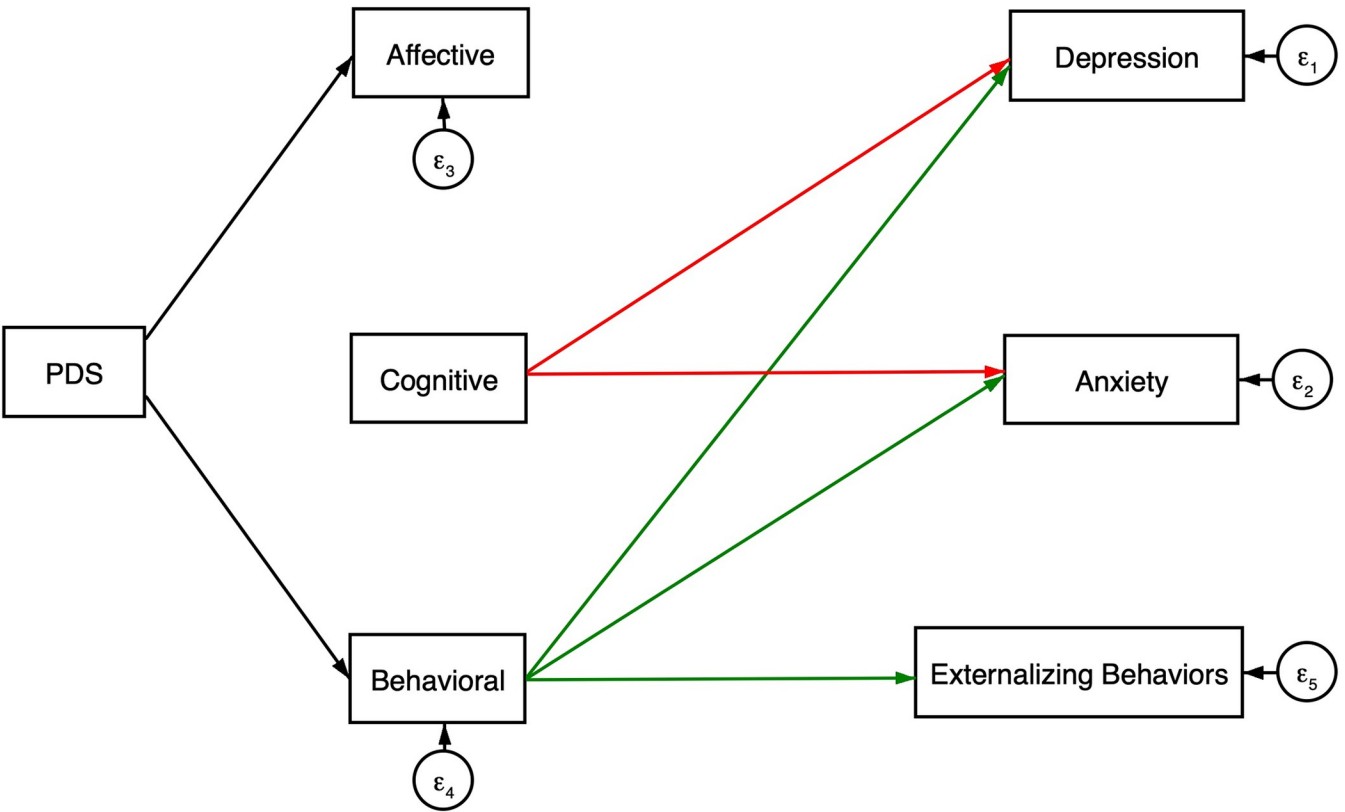

**Fig 2. Significant structural paths between pubertal development stage, empathy dimension and psychological symptoms.** Note. Green paths are associated with lower psychological symptoms; Red paths are associated with higher psychological symptoms; Black path arrows indicate positive correlation.

distress of others, without having the skills necessary to intentionally alleviate this distress by effectively comforting others may result in the tendency to co-ruminate [14]. A study on rumination as a transdiagnostic factor in anxiety and depression by McLaughlin & Nolen-Hoeksema, 2011, found that rumination was a more significant mediator in the co-occurrence of symptoms of depression and anxiety in adolescents than adults [49]. McLaughlin & Nolen-Hoeksema also noted no findings of rumination as a mediator of anxiety and later increases in depressive symptoms among adolescents, contrasting other prospective studies [49]. They hypothesized this discrepancy may have resulted from differences in the demographics of the adolescent samples between studies, with their study sampling more economically disadvantaged and minority racial/ethnic groups [49]. These differences highlight the need not only for longitudinal data to examine trajectories of development and change in these relationships, but also the importance of comparing these results across samples of adolescents from diverse cultural and socioeconomic backgrounds to understand how and when cognitive empathy and rumination impact mental health.

After examining sex, age and pubertal development status in relation to empathy dimensions and outcomes, it was determined that PDS, but not age or sex were significantly associated with both empathy dimensions and psychological symptoms in this sample. The failure of age to be significantly associated with empathy and psychological symptoms may be partially attributable to the variation in the age in onset of puberty, particularly between different genders. Sex, while associated with affective empathy, was not associated with the primary outcomes after controlling for PDS and age. While evidence supports gender differences in

mental health outcomes, it is likely that the narrow age range of early adolescents included in this sample represents early development, prior to differentiation in mental health outcomes that widen during mid to later adolescence. This finding further underscores the importance of considering pubertal development stage, more than age and gender when assessing empathy and psychological symptoms in very young adolescents. Moreover, in this sample the interaction term between PDS and higher cognitive empathy was significant and indicates the important developmental association with risk for psychopathology.

Behavioral empathy, and the acquisition of skills to practice empathy through intentional action to comfort others was associated with lower depressive, anxiety and externalizing symptoms. Interventions targeting behavioral empathy should consider tailoring approaches and content to specific developmental stages, such as early adolescence, to protect against risk for psychopathology before vulnerability increases during mid to later adolescence [9]. Behavioral empathy and effectively comforting others are skills that require experiential practice to master and younger adolescents may have fewer opportunities than older adolescents or adults to practice these skills. Acquisition of behavioral empathy skills also holds potential to promote wellbeing such as increasing friendship quality, prosocial behaviors and social support systems during a critical developmental period [10–12]. Interventions should consider duration and frequency of interventions targeting empathetic capacity, because researchers have hypothesized that cognitive components of empathy may require longer, sustained interventions than affective components [50]. Research is needed to consider the duration and frequency needed to support behavioral empathy, a newer dimension of empathy that our results indicate are associated with lower internalizing and externalizing symptoms.

Previous research notes that there can be large variations in empathy during development, even within a particular grade level, which can greatly impact the effectiveness of an intervention [50]. To generate developmentally informed intervention approaches it is important to move beyond age as a proxy for development. Use of early screening and developmental assessment tools can improve program planning. Early adolescents ages 10–14, have wide variation in pubertal development stage as well, especially between sexes, with girls transitioning through puberty earlier than boys on average [51, 52]. This further emphasizes the importance of a tailored approach that aligns interventions to developmental stage and promotes appropriate skills and approaches that fit within the zone of proximal development.

More research is needed to evaluate how social emotional learning interventions target or affect affective, behavioral and cognitive empathy dimensions. In addition research should evaluate possible associations between cognitive empathy and co-rumination. Given the promising protective paths from behavioral empathy to depression, anxiety and externalizing behavior, this study suggests that behavioral empathy may be especially important to support during early adolescence.

## Limitations

This study was limited to cross-sectional data. Longitudinal data is needed to identify causal relationships in empathy trajectories across development. Longitudinal research should also consider potential differential impacts of socioeconomic status and cultural factors on the associations between measures such as empathy, pubertal development stage, rumination, and psychological symptoms and mental health. Additionally, the measures of behavioral empathy captured by the "intent to comfort" subscale of the EmQue-CA does not capture actual actions that the young person performed. Further research to assess what kinds of behaviors are appropriate for early adolescents to act on and that provide protection from mental health disorders are important to identify. We recommend additional experimental and longitudinal

research to better assess developmental trajectories in causal relationships of empathetic dimensions and psychological outcomes.

## Conclusion

These findings support the importance of examining developmental stage during early adolescence over age in testing predictors of mental health outcomes. Programs targeting empathy should consider the components of the intervention and whether they target affective, cognitive, or behavioral dimensions. To strengthen the translation of research to practice it is important to identify dimensions of empathy as mechanisms of change, and implications for mental wellbeing.

## Supporting information

**S1 Dataset. Empathy Tanzania year 3 timepoint 1 data.** Datafile used for statistical analysis. (DTA)

## Acknowledgments

The authors would like to acknowledge Health for a Prosperous Nation (HPON) for their work in recruiting study participants, obtaining consent and assent and administering the survey questionnaire. Special thanks to the Discover National Advisory Board Members, Ministry of Education and Ministry of Health for supporting this work. The authors are grateful to the National Institute of Medical Research in Tanzania (NIMRI) and the University of California Berkeley Committee for Protection of Human Subjects Institutional Review Board for guidance and approval of the study. Most of all the authors would like to thank the adolescent participants for their participation in the study and their parents/caregivers for supporting participation in the study.

## Author Contributions

**Conceptualization:** Megan Cherewick, Prosper Njau, Ronald E. Dahl.

**Data curation:** Emily Hipp.

**Formal analysis:** Megan Cherewick, Sarah Schmiege.

**Funding acquisition:** Ronald E. Dahl.

**Investigation:** Megan Cherewick.

**Methodology:** Megan Cherewick, Jenn Leiferman.

**Project administration:** Megan Cherewick, Prosper Njau, Ronald E. Dahl.

**Supervision:** Prosper Njau, Ronald E. Dahl.

**Writing – original draft:** Megan Cherewick, Sarah Schmiege, Emily Hipp, Jenn Leiferman, Prosper Njau, Ronald E. Dahl.

**Writing – review & editing:** Megan Cherewick, Sarah Schmiege, Emily Hipp, Jenn Leiferman, Prosper Njau, Ronald E. Dahl.

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
