## [Decision Letter · Decision Letter 0]

19 Sep 2022

PGPH-D-22-01279

A developmental analysis of dimensions of empathy during adolescence: Behavioral empathy but not cognitive empathy is associated with lower psychopathology

Dear Dr. Cherewick,

Thank you for submitting your manuscript to PLOS Global Public Health. After careful consideration, we feel that it has merit but does not fully meet PLOS Global Public Health’s publication criteria as it currently stands. Therefore, we invite you to submit a revised version of the manuscript that addresses the points raised by the reviewers during the review process.

We look forward to receiving your revised manuscript.

Kind regards,

Rakesh Singh

Academic Editor

Journal Requirements:

a. State what role the funders took in the study. If the funders had no role in your study, please state: “The funders had no role in study design, data collection and analysis, decision to publish, or preparation of the manuscript.”

Reviewers' comments:

Reviewer #1: On the intorduction:

1. In lines 72-73 about studies showing association of empathy and prosocial behavior and externalizing problems, it would be interesting to see a slightly more detailed explanation as to what kind of empathy and in what population the results were found, so that the potential of the presented paper is clearer (do we need to understand how it works in a different population such as young adolescents in Tanzania, etc.)

2. In lines 104-107 on the brain structure related to empathy, it could help to explain it in lay language so that the non expert audience can understand easily.

On materials and methods:

1. In lines 135-136 the protocol of the parent study Discover Learning is mentioned and it's indicated that it's provided elsewhere, but it would be interesting to include at least a brief explanation of what it entails and why it is used.

2. It's not clear why only verbal assent from adolescents was required, they could also provide written assent unless explained why (culture, literacy level, etc.)

3. It is mentioned that each measurement scale was selected based on previous use in low- and middle-income countries with adolescent populations and that items were adapted to reflect the study context and age of participants. Howev, it is not clear what use they gave them in other studies and how the adaptation was done for this study, e.g. it's not explained how the PDS questionnaire was validated in Tanzania, and for AYPA the developmentscale is mentioned but not for the target population of this study.

4. The poverty profile of the participants is mentioned but not really analyzed.

On results:

1. It looks like something was incorrectly written in line 240, it states: "with cognitive empathy anxiety", probably "anxiety" was incorrectly included.

2. The robustness of the multivariable regression is not clearly explained, if some tests were performed, it would be relevant to show them. For example, if the different empathy dimensions are correlated, or age and PDS are correlated, wouldn't this be presenting problems of multicollinearity?

On discussion:

1. The writing in lines 304-306 seems repetitive.

2. The studies of rumination and co-rumination suddenly appear here, but they were never mentioned as a fraework or basis for the analysis in the introduction. It would be clearer if the concept is introduced in the introduction so that it makes more sense to bring it up in the discussion and analysis of results.

3. In line 327 is is mentioned that an objective was to test sex (in addition to age and PDS) in relation to empathy dimensions and outcomes, but the sex variable is then never analyzed. If there were no relevant results, it would be worth mentioning it.

4. Lines 351-352: what about behavioral components? The previous consideration was on behavior but it’s not mentioned here for the intervention duration.

5. The general discussion of the paper on the relevance of generating developmentally informed intervention approaches is very convincing given the results and presents an important contribution to the area of study.

On limitations: The limitations are clear and well thought to move forward in the field.

Reviewer #2: This manuscript covers a topic of interest in the fields of developmental psychology and global mental health.

The authors of the manuscript may want to provide more detail on their definition of externalizing behaviors so readers have a better sense of the how cognitive and behavioral empathy relate to certain behaviors.

The alpha level and a sample question from the Pubertal Development Scale should also be reported.

The discussion section contains sound recommendations for future studies given the results of the current study.

It is recommended that the authors review the manuscript for minor typos to ensure that ideas are communicated clearly.

---

## [Editor Report · Decision Letter 1]

12 Oct 2022

A developmental analysis of dimensions of empathy during early adolescence: Behavioral empathy but not cognitive empathy is associated with lower psychopathology

PGPH-D-22-01279R1

Dear Dr. Cherewick,

We are pleased to inform you that your manuscript 'A developmental analysis of dimensions of empathy during early adolescence: Behavioral empathy but not cognitive empathy is associated with lower psychopathology' has been provisionally accepted for publication in PLOS Global Public Health.

Best regards,

Rakesh Singh

Academic Editor
